

# Automated observatory in Antarctica: real-time data transfer on constrained networks in practice.

Stephan Bracke[1], Alexandre Gonsette[1], Jean Rasson[1], Antoine Poncelet[1], Olivier Hendrickx[1]

[1] Institut Royal Météorologique (IRM), Centre de Physique du Globe, 5670 Viroinval (Dourbes), Belgium.

*Correspondence to*: Stephan Bracke (stephan.bracke@meteo.be)

**Abstract.** In 2013 the scientists from the geophysical centre at Dourbes started a project to install a fully automated magnetic observatory in Antarctica. This isolated place comes with specific requirements: unmanned station during six months, low temperatures with extreme values up to - 50 °, minimize power consumption, satellite bandwidth limited to 56 Kbit/sec. The ultimate aim is to collect real-time magnetic data every second in Belgium: vector data from a lemi-25,

absolute F measurements from a gems proton magnetometer and absolute magnetic inclination and declination measurements (approximately 5 times a day) with an automated DI-flux. To achieve real-time data transfer with traditional file transfer protocols (for instance sftp, mail, rsync), you fight against their limitations in terms of real-time. After evaluation of pro and cons of the on that moment available real-time internet of things (IoT) protocols and seismic software solutions known to UGCS, we chose to use message queuing telemetry transport (MQTT) and receive the one second data

with a negligible latency cost and no loss of data. Each individual instrument sends his data towards Belgium immediately after capturing it and arrives at approximately 300 milliseconds after sending which corresponds with the normal satellite latency.

## 1 Introduction

Princess Elisabeth Antarctica (Figure 1 : right PEA), located on Utsteinen Nunatak in Queen Maud Land (71.949960°S

23.347503°E), is a Belgian scientific polar research station, which went into service on February 15, 2009. It is approximately 220 km from the Antarctic coast, which makes it an ideal logistics hub for field exploration in the 20°- 30° E sector of Antarctica. The station is unmanned during the Antarctic winter period but each year from November until February there is a crew that makes it possible for scientists to plan research in this remote and isolated place. In February 2014 a first mission was planned by the magnetic scientists of Belgium to locate a place where we could install a magnetic

observatory nearby the Princess Elisabeth station where we can still profit of the infrastructure of the station, but remote enough to reduce any magnetic interference that could be caused by activities around the station. This resulted in finding the best spot to create a non-magnetic radome at approximately 500 meters of the station (Figure 1: left). The location is near to the Utsteinen mountain to make it possible to fix the radome on solid rocks and by doing so guaranteeing that the pillars will



not move. The radome has some limitations to be taken into account: it is not heated, power supply will be delivered from the station but consumption should be minimized as much as possible.

During the season of 2014-2015 a second mission took place. The main goals were to install the first two instruments for continuous monitoring of the magnetic field and establish a way to communicate the data back to Belgium. First instrument is a vector instrument which measures continuously the variation of the magnetic vector field around three axes (XYZ). We chose for the LEMI-25 (Figure 2 : left) for its known temperature stability. Because the radome is not heated the LEMI-25 got adapted so that it could withstand the extreme cold temperatures. The second instrument is a scalar instrument that measures continuously the magnetic field strength. For this we installed a GEMS GSM-90 (Figure 2: right). Both instruments have a sampling rate of 1Hz and come with a windows program that constantly logs the measurements in a file. Because of the limitations of the radome we opted not to use full blown windows pc's but looked into the possibilities of small ARM processors. Thanks to popularity of Rasberry PI these small cheap ARM processors gain in popularity and opened up a whole new market. The one that got our attention is the Beaglebone Black (Figure 3): a small Texas Instruments single board computer. It is comparable with the Raspberry but unlike the latter it is more designed for robotics having the following advantages:

- The hardware is completely open-sourced so adaptations or own productions are fairly easy.
- A rugged version for extreme temperature is available.
- It comes with two 46 pin GPIO headers
- It has on board 4 Gigabyte of internal storage making the boot from SD card not necessary.

As these boards are linux based computers the needed software has been rewritten and both instruments are controllable via a web-interface. Having established the possibility to install and collect the data for these instruments in the radome without violating the requirements, there is still one main question to solve: How to transfer the data in real-time to Belgium? As we can see on the schematics depicted in Figure 4 the radome got connected to the Princess Elisabeth station with fiber-optics. Princess Elisabeth station has a permanent satellite communication and limits each scientific project to 56 kbit/s communication to the base station in London. From there on it is available on the internet. We will look in the next chapters how we tackled this situation to an attempt to realise real time data transfer on a satellite link without losing data.

## 2 Towards real time data transfer

To be able to talk about real time it is necessary that we come to a mutual agreement of what "real-time" exactly means. For this we can take the following definition of real time:

Real-time or real time is a term often used to distinguish reporting, depicting, or reacting to events at the same rate and sometimes at the same time as they unfold, rather than compressing a depiction or delaying a report or action. The term has uses in many areas, and many articles cover different aspects.

On the other hand in terms of computers collecting data and transmitting it we often fall back to near real time data meaning:





The term "near real-time" or "nearly real-time" (NRT), in telecommunications and computing, refers to the time delay introduced, by automated data processing or network transmission.

So applying this to our example of our two instruments who collect data every second and communicating it to our ARM computer using a serial port introduces already a delay (for the lemi-25 there is a delay of 0.3 s only to combine signal with GPS). Taken into account that the seconds data of the LEMI-25 are a calculated Gaussian mean of 10 Hz samples, we introduce a new delay only to be able to calculate these values. With this in mind, we can state that real-time in the pure sense of the word will not be possible and to be correct we should always talk about near real time data. It is now our challenge to limit the time that it takes to send the data from the data capturing on the serial port over the network via satellite and the internet towards our servers in Belgium.

## 2.1 Traditional ways of data transfer

When we look at the way data transfer is often done in the magnetic data collection projects we mostly fall back to file transfer. Most of the time data is written to a file and this one is then transmitted to Belgium. Popular solutions are

- Rsync : is a linux utility that synchronizes directories over the network. Rsync typically uses SSH connections to encrypt and secure data transfer. It will only update the necessary differences.
- FTP: Classical file transfer protocol that makes it able to copy files to a remote server. When security and encryption is needed FTP can be extended with certificates ( FTPS)
- SFTP: same as FTP but underlying connection is different. The security is done over SSH as in rsync which comes with a latency impact compared to the pure FTP(S) but is simpler to use behind firewalls and highly secured.
- Mail: traditional mail can be used to send the file in attachment to a destination. As it is mail there is always an uncertainty of delivery and an unknown latency.

All these protocols have their value as being standard and easy to use but in terms of near real time they add a serious overhead. First of all the files need to be created and secondly the protocols are focussed on sending a file correctly but not really on fast transfer with limited overhead. These protocols can attain near real-time in terms of minute delay, but not if we aim to reduce the delay to seconds.

## 2.2 Alternative approaches

As previous protocols where all based on file transfer, we should shift our focus on data transfer protocols to find an answer to providing real time data availability. If we need to find solutions on the best protocols available, we can be inspired by colleague scientists or just go and look what the open source world has to offer.



### 2.2.1 Protocols used in seismology

Scientists who study the earth magnetic field often have a shared knowledge with seismologists. In seismology real-time data transfer was always an important requirement. At USGS two software packages are mentioned that promise real-time data transfer: earthworm and antelope (proprietary). A third one that has gained popularity is called SeisComP3. These three

packages are offering more than just data transfer and often come with seismologic data visualization and analysis tools. They are basically full blown server implementations (with tools and database to configure) and although they have different possibilities to communicate data, each of them comes with the possibility to send data to it in the form of SeedLink packages a data transfer protocol defined by the incorporated research institutions for seismology (IRIS). As stated on their website: The SeedLink protocol is a robust data transmission intended for use on the internet or private circuits that support

TCP/IP. The protocol is robust in that clients may disconnect and reconnect without losing data, in other words transmissions may be resumed as long as the data still exist in the servers buffer. Requested data streams may be limited to specific networks, stations, locations and/or channels. All data packets are 512-byte mini-SEED records.

Although data packets of 512 bytes are called mini-SEED records in the world of 1 second magnetic data, which can be packaged in +/- 30 bytes, these are big. When it comes to limited bandwidth connections on satellite links, sending packages

that are only filled with 10 % of useful data becomes a waste of resources. Secondly it is not so straight forward to install one of the previous mentioned servers (needed to collect the data) and comes with a lot of overkill if they are not used for seismologic data. As a conclusion we can state that using these kinds of solutions to transfer magnetic data is a valuable option if on your servers you have already earthworm, antelope or SeisComP3. If not it will come with a big learning curve and sometimes poorly documented support.

### 20 2.2.1 The internet of things.

Today we live in a world where everything is connected and all devices become "smart". The days were there were only one or two computers in a family are definitely over. Devices registering data and sending data over the internet: tv's, smart phones, smart thermostats, smart lights, surveillance systems, robotic lawn mowers, etc. Millions of small devices with sensors send in continuous mode data over the internet. Thanks to this explosion in commercial applications we as scientist

can profit of the evolution of data transfer protocols to apply these emerging needs. The application often comes with similar requirements as the ones we have for instruments on remote locations:

- Need to run on small devices
- Minimize power usage ( often devices or on battery )
- Run on unreliable networks ( Wi-Fi, mobile networks )
- Send regularly small messages.



Another advantage of the software community is that there is a large open source community driven by sharing knowledge and solving problems together. On this base a search on the internet on data transfer protocol in December 2014 resulted in three protocols that were suitable to establish this real time data transfer:

- Advanced message queueing protocol ( AMQP )
- Streaming text oriented protocol ( STOMP )
- Message queue telemetry transport ( MQTT )

First of all these three mentioned protocols are not implementations, they are just program language-agnostic descriptions how clients and server can communicate on an asynchronous level. Two of them: MQTT and AMQP are even evolved into an OASIS open standard, with the advantage that different implementations of client libraries and servers have emerged in

both open source and commercial landscape. To be able to choose among these protocols it is important to look at their specifications and their targeted audience. While taken the definition from there respectively website:

AMQP was designed as a replacement for existing proprietary messaging middleware (IBM, Microsoft). It is a full blown business to business message protocol with focus on reliability, interoperability and security. The protocol makes it possible to apply flexible routing of messages, with transaction support. The protocol is at his lowest level an efficient, binary, peer-

to-peer protocol for transporting messages between two processes over a tcp/ip connection.

MQTT is a machine-to-machine (M2M)/"Internet of Things" connectivity protocol. It was designed as an extremely lightweight publish/subscribe messaging transport. It is useful for connections with remote locations where a small code footprint is required and/or network bandwidth is at a premium. It is a binary protocol over tcp/ip.

STOMP is a protocol that is completely text based making it more in favour to be used over HTTP. It stays very simple and

has no knowledge of transactional context meaning that there is no direct support for any guarantees of delivery.

In following table we summarized the most important criteria for my application purposes.

| protocol | lightweight | binary | Quality of Service | Designed for low-bandwidth networks |
|----------|-------------|--------|--------------------|-------------------------------------|
| AMQP | No | Yes | Yes | No |
| MQTT | Yes | Yes | Yes | Yes |
| STOMP | Yes | No | Not out of the box | No |

From this table it is clear that among these three protocols MQTT is the most appropriate to solve the data transport for our

installation in Antarctica

## 3 MQTT explained

Message Queue Telemetry transport is not new. It was designed and documented by Andy Stanford-Clark and Arlen Nipper of Cirrus Link Solutions (a company specialized in real-time telemetry solutions) in 1999. The design principles were to



minimize network bandwidth and device resource requirements whilst also attempting to ensure reliability and some degree of assurance of delivery. In 2011 it was even used by Facebook to reduce their phone-to-phone delivery of the messenger from several seconds to hundredths of seconds. What makes it interesting for us is that in 2013 IBM submitted MQTTv3.1 to the organization for the advancement of structured information standards (OASIS) making it available for everybody to use

and implement it into their applications. From that moment on mqtt has definitely evolved to one of the most used protocols in the internet of things applications.

### 3.1 Publisher and Subscribers

The mqtt protocol is basically a publish/subscribe mechanism which needs a broker (Figure 5) to communicate between those who send messages (Publishers) and those who receive messages (Subscribers). Publishers and subscribers are

completely decoupled and aren't depending on each other meaning that a publisher doesn't need to be up and running to be guaranteed to receive messages. For each message published there can be multiple subscribers to receive the messages. We could compare it to subscription to a newsletter. Messages are published on topics. In mqtt topics are UTF-8 strings: for example **myhome/floor/room/temperature**. The publisher here on this topic will be a small device with a temperature sensor which reads on regular intervals a temperature and publishes this on the topic. The messages are byte arrays so it's up

to the programmer to clearly define the message and document it so subscribers can interpret the message. From now on anyone who needs the temperature can subscribe to the topic and will receive the byte array. It is obvious that publisher and subscribers are most of the time programs. To make mqtt work we can use client libraries which are available in multiple programming languages. On the main MQTT site you can find over 60 different client libraries going from shell scripts over device specific ones like the one for Arduino to the most popular programming languages. Applying this to our problem we

adapted the programs on the beaglebones (Figure 3) to be able to publish the magnetic data immediately after it is read from the serial port communication with the device. This enabled us to publish data captured  in the radome on Antarctica directly to a broker installed on a server in Belgium. As topic structure we took the following approach:
iagacode/instrumentid/samplerate : pea/lemi0001/sec meaning that this topic receives 1 second vector data ( lemi = vector instrument) located at Princess Elisabeth Station Antarctica (pea) This topic is also linked with a description of the message

structure posted on it, so that from now on we can listen to this topic and receive near real time data coming from Antarctica

### 3.2 The mqtt broker

The client libraries for Publisher and Subscribers are lightweight, but before the can communicate with each other you need another piece of software called the broker. Also here you have lots of them available (some open-sourced, some proprietary). The choice of the correct one will depend on the load you want to send to it and the ability these server can

cope with this. As from our application the load is very low and so we chose for the most popular, lightweight and easy to use open sourced one called mosquito.


His main responsibilities are:

- Make topics available for publishers and subscribers
- Ensure receiving and correct delivery of the mqtt byte messages
- Configure security

## 3.3 Quality of service

As the foot print of the mqtt is tiny and overhead small it makes it also fast. Although this is Ok there is another thing that is important on unreliable networks: what the possibility of losing messages during transfer is? For this mqtt introduces three levels of quality of service (QOS).

- QOS 0: lowest level of assurance. The message will be send at most once, but it will not survive failures. It will be doing so never introduce duplicates. Often it is referred to as "fire and forget".
- QOS 1: The message will be send at least once. This quality of service introduces the possibility of duplicate messages (it is the subscriber that can receive messages more than once). This QOS will survive connection loss. This however requires an acknowledgement back from the server before the client can discard the message.
- QOS 2: The message is send exactly once. The subscriber will be guaranteed to receive the message exactly one time. This QOS survives connection loss but introduces extra layer of communication messages between publisher and broker ( two extra messages to assure that message was received )

QOS 0 is by all means the fastest and the least band-width consuming. However acknowledge messages are 1 byte so the overhead compared with the disadvantage of being sure the message arrived is negligible and often (as in our case) the preferred way of sending messages. Take into account that QOS can be set on each individual messages, so it is simple to change QOS between messages.

## 4. Experiences after one year of data flow

With this in mind the setup was done in February 2015. On level of mqtt we took the following decisions
Two publishers send data:

- Vector instrument lemi-25 sends each second a byte messages of 16 bytes
- Scalar instrument gems-gsm 90 sends each second a  messages of 8 bytes
- Both use a quality of service 1 which guarantees that messages arrive at least once

In our situation we make an mqtt connection and we never close it. The library used (in our case nodejs library) will automatically reconnect if connection gets lost. To establish this we need to take into account another important parameter called keep alive. The keep alive interval is the longest possible period of time, which broker and client can endure without sending a message. If broker or clients don't receive any message during this period the connection will be closed and client



will try to re-establish the connection. We have set the keep alive to 10 seconds. On protocol level this means that as long as we assume there is a connection we can send messages and the connection can be broken without noticing it for maximum 10 seconds. On this unnoticed stale connection, we could continue to send messages of QOS1 that will never be acknowledged by the broker (maximum 10 because of the keep alive which assures to kill the stale connection after 10

seconds). After the connection is re-established these messages will be send automatically again by mqtt because of the assurance of QOS 1. If we however try to send a message during the time we have no connection (for example a physical problem on the satellite itself or server with the broker on it) the library will respond with an error and it is our job to deal with this not send messages. To cope with this problem we just used a memory queue which stores 4 hours of second data. This means that from now on we can live with connection failures up to 4 hours (keep in mind that we lost our real time

promises here).

So during one year data transmission we evaluated that the mean delivery time is 300 ms which corresponds with standard satellite delays (no real impact due to the mqtt protocol). Connections got lost +/- 10 times a month (small failures of a couple of seconds), these are re-established and data is resend without any problem. During this period we had only one big connection lost mainly because of the fact that a fibre optic cable at Belgium got broken. It took two days to recover the

connection. To recover these two days of data we felt back to standard ftp of the recorded files at Antarctica.

## 5. Conclusions

Realizing near real-time data transfer today is feasible with standard open protocols and open source tools. It doesn't come for free because it introduces the need of managing and monitoring a message broker. All research was done in 2014, when we re-evaluate the taken decisions we can see that today mqtt has evolved to a mature near real time data transfer protocol

widely adopted. Although we can't neglect that other new promising alternatives need to be investigated:

- MQTT-SN: MQTT for Sensor Networks is aimed at embedded devices on non-TCP/IP networks, such as Zigbee. MQTT-SN is a publish/subscribe messaging protocol for wireless sensor networks (WSN), with the aim of extending the MQTT protocol beyond the reach of TCP/IP infrastructure for Sensor and Actuator solution
- CoAP: The Constrained Application Protocol (CoAP) is a specialized web transfer protocol for use with constrained

nodes and constrained networks in the Internet of Things. The protocol is designed for machine-to-machine (M2M) applications such as smart energy and building automation.

While this was realized during the month of February 2015 a third mission needs to be planned in the next season to install an automated DIFlux called a GyroDif. This will eventually result in the realisation in the first fully automated magnetic observatory with near real time data transfer



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

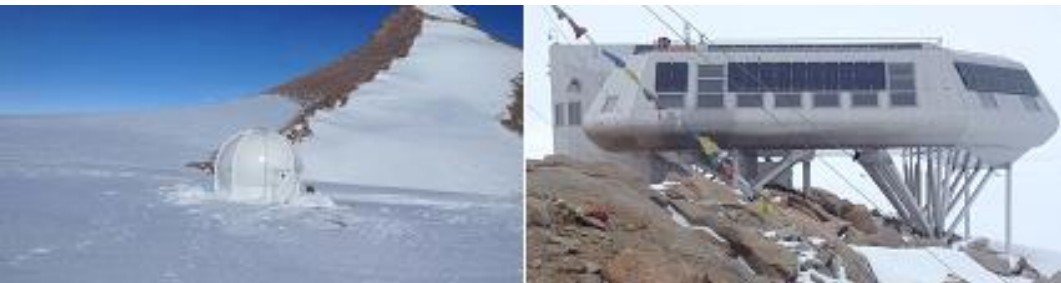

**Figure 1: Magnetic radome and Princess Elisabeth Station**

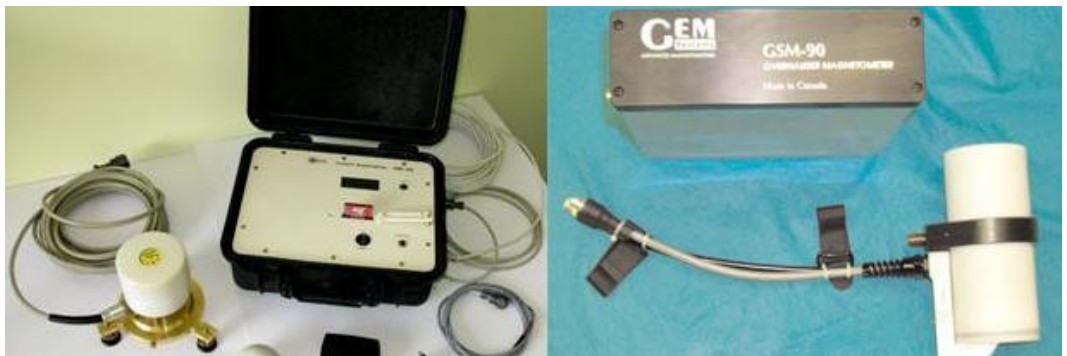

**Figure 2: LEMI-25 (left) GEMS GSM-90 (right)**

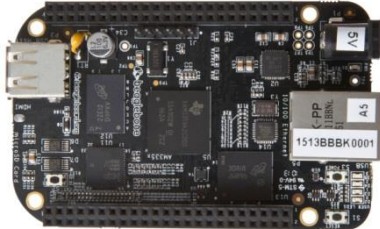

15

**Figure 3: Beaglebone black ARM processor**





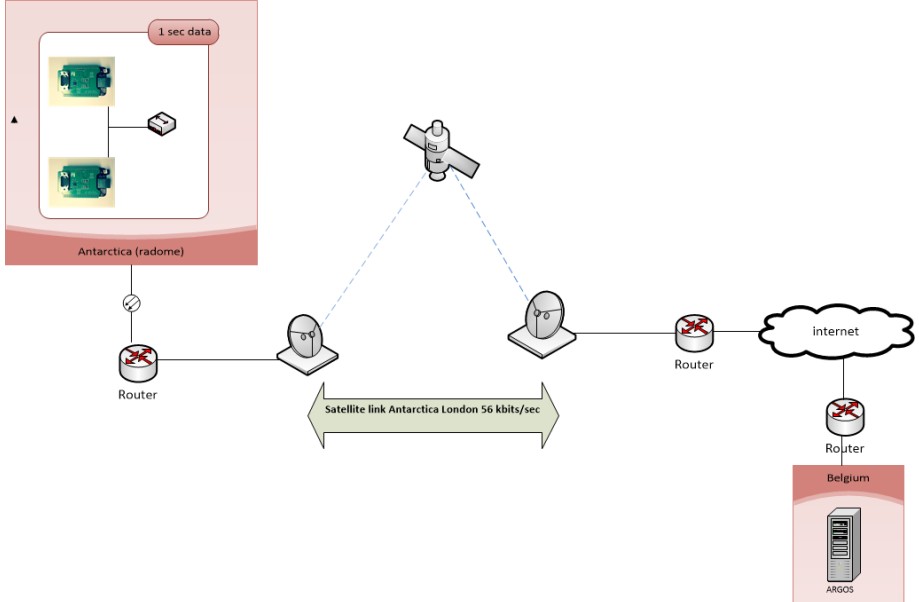

**Figure 4: data transfer problem**

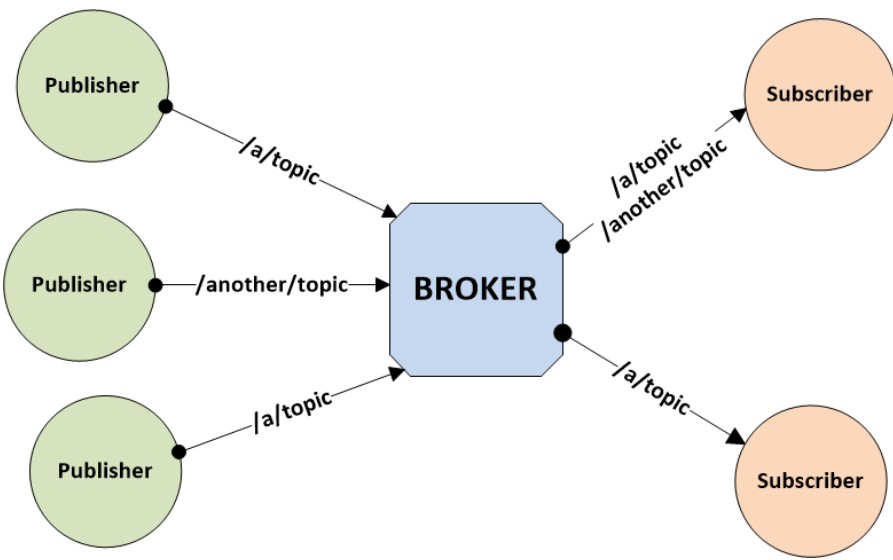

**Figure 5: MQTT broker**