# Peer review of "Automated observatory in Antarctica: real-time data transfer on constrained networks in practice."

_Geoscientific Instrumentation, Methods and Data Systems, 2017_

## Referee Comment (RC1) · C. Blais (Referee) · 27 Mar 2017

The paper is good but it would be interesting to see a more detailed example of the implemented work. From understanding your infrastructure limitations at the remote observatories, it is understandable that any file transfer protocol (RSYNC, FTP, etc.) are far too demanding on bandwidth. The Beaglebone is definitely interesting piece of hardware and MQTT does look like a promising alternative to data transfer. It would be interesting to briefly mention why Belgium requires real-time data. Is it for emergency management?

On the section "Traditional ways of data transfer", it might be good to note that none of these protocols are meant for real-time data transfer. To say that these protocols

can attain near real-time in terms of minute delay is against their intended purpose. RSYNC, for one, is meant for archival purposes. It might be good to simply restate that all these protocols are bandwidth intensive. Even RSYNC needs to send content to determine what needs to be updated.

On the section "Protocol used in seismology", Gempa's SeisComP3 has to be ability to receive SeedLink but also has CAPS. CAPS is their own documented real-time acquisition server with additional capabilities of synchronizing gaps from other data sources; for example a backup data center. Their protocol allows safe duplication in disaster recovery solutions with relatively small messages. CAPS server can also be purchased at reasonable cost and with very good support. As for IRIS SeedLink server (ringserver), it doesn't require much effort to install as it can be done with a single "make" command. As a note, miniSeed is not limited to seismological data as many other institutes use it for other non-seismological instruments. In fact, SEED did reserve a code for magnetometer (F). miniSeed supports almost any type of fixed sampling rate data. In other words, SeedLink is not reserved for seismologic data. FDSN, on other hand, is very seismological centric but they do not limit anyone from getting a network code for miniSeed (if you ever intend to make miniSeed public). However, available network codes are limited.

On the section "Experience after one year of data flow", it is not clearly explained so I will assume your MQTT broker is located at the data center. This means that the remote station must make the connection to the data center and must have the data center IP hard-coded on site. What happens if the data center IP changes (ex: disaster recovery)? The data center is the most accessible system; wouldn't it be easier to initiate connections from there (no hard coding at remote stations)? Also, the purpose of the MQTT broker is to buffer messages if connections are lost. In your case, you implemented your own buffer on site. What is the purpose of the MQTT buffer at the data center?

On the section "Conclusion", it might be good to evaluate ActiveMQ (message broker). In seismology, ShakeAlert is a complex system where it exchanges messages between internal modules using ActiveMQ. This system will eventually alert the public in real-time about incoming earthquakes. Clients will get the alerts by connecting to an ActiveMQ broker located at a warning center.

---

## Short Comment (SC1) · 10 Apr 2017

Why does Belgium require real-time data ?

Belgium is not directly in need of real-time data although there are some domains where we see possible usages : The real time monitoring of the ionosphere done by our ionospheric section, the demand to test real-time solutions for intermagnet and the guidance of directional drilling done on oil plants. The installation of a magnetic observatory on a remote location was for me the ideal occasion to do some tests in this direction. I can add this to the paper.

I will add the note you mentioned on "Traditional ways of data transfer"

[Figure]

Where did I install the MQTT broker? On level on choosing the place of the MQTT broker it basically falls down in three possibilities: On the remote location, on the data centre site or by bridging two brokers ( on both sites brokers where we bridge topics) My choice to put it on the data centre site is basically because I had no control on firewall settings on the remote location and this gave me the possibility to configure everything correctly before arriving on site.( I can add this to the paper) Your remark on IP addresses and change of IP address was not valid for my setup because it is not on the ip addresses of big data servers. The mqtt broker (in my case) is very lightweight and has its own public reserved ip address, so it is very easy to recreate once the server is not operational.

Question on buffering on sender itself? It is always valid of having a minimal buffer on sender site. This makes it completely independent of the broker. If you upgrade broker, reboot server where broker is installed on, etc. the connection will be lost and if sender buffering is implemented the impact will be minimal. The server indeed buffers messages but this is only valid for guaranty of delivery to the receiver that doesn't need to be online all the time.

Evaluation of ActiveMQ ? The mentioning and evaluation of different brokers is out of scope for this paper. There are many out there and each with their strength and weaknesses; the advantage of MQTT is that you're not bound to one broker and can choose in function of your knowledge and needs. If you mention ActiveMQ, it can be appropriate to look at Apollo ( https://activemq.apache.org/apollo/ ) which is the next generation of ActiveMQ and tends to scale better in terms of mqtt .

Remarks on protocols used in seismology:

I will delete my second argument as I state that it is not straight forward to install and basically restate that code samples and ways of usages are difficult to find, compared to the IoT solutions. I have to mention that I had two months to select, implement and test a solution, beside that I needed lots of different client implementations: C++, java,

python, nodejs (javascript) and C#, I needed well documented solutions with a large community and that is where open source IoT solutions excel

---

## Referee Comment (RC2) · R. Leonhardt (Referee) · 13 Apr 2017

Review of "Automated observatory in Antarctica:..." by Bracke and other.

The manuscript presents a case study on setup and , particularly, near-real-time data transfer from remote locations. Different protocols are discussed for the latter. The manuscript deals with a frequently discussed issue in geomagnetism. I very much welcome this contribution. It is well structured and written. I strongly recommend acceptance of this manuscript. Nonetheless, I have a few points which I would like the authors to consider or comment on:

Specific aspects:

[Figure]

1.) Discussions on data protocols: Most aspects have already been addressed by the other review and the reply. Just one comment on miniSeed: as you intend to broadcast DI data from your automatic Di-Flux as well, protocols which require evenly spaced time intervals as miniSeed are not favorable. In section 4, last sentence, you mention that you can use non-real time data transfer as well. It would be good to add this aspect earlier where you describe your system layout and its connections. Having a fall back is definitely worth to mention. Is this working automatically, i.e. running if a certain amount of data is missing?

2.) Description of the observatory: It would be great if you would add a better description of the observatory itself. You just show two photos of the the dome and the station. A map or sketch indicating the positions of the instruments, electronics, data transfer connections, position of broker, would be very welcome.

3.) Please add references to all citations (e.g. page 2, ll 32) and the descriptions of the protocols.

Other comments:

1.) Consistent use of abbreviations: I recommend use capitals for MQTT, GEMS, LEMI throughout the manuscript. In any case don't mix.

2.) Page 2, line 7: LEMI got adapted: It would be good if you could briefly explain what you mean by "adapted".

3.) Page 2, line 5: "First instrument ... . We chose ..." -> "For measurements of the magnetic field vector we chose ..."

3.) page 8, line 4: "We have therefore hypothesis..." -> "Our hypothesis is ..."

4.) As your article is titled "data transfer ... in practice" you could provide some more info on the transmitted data itself. Some descriptive graphs on connection failures (only about 10 times each month), and delayed transmission would strengthen section 4.

Best wishes, Roman

---

## Author Comment (AC2) · 21 Apr 2017

Remark on miniSeed intervals : Indeed for DIFlux measurements evenly spaced time intervals are not appropriate, so using miniseed for this is not recommended. I will mention this in the paper.

Remark on using non-realtime file transfer as fallback mechanism. It is important to guarantee completeness of data. For the moment it is not fully automated ( only warnings are created when data is missing after one day) but the aim is to automate this in the near future. I will specify it in the paper.

Remark 2 and 3 : description of the observatory and references will be added.

[Figure]

Remark on "Lemi got Adapted" The lemi got adapted for low temperature by the people of LVIV Centre of Institute for Space Research on our special request.I never was informed on what they did to achieve this.

Other comments : All will be taken into account

---

## Referee Comment (RC3) · Anonymous Referee #3 · 24 Apr 2017

**1  General remarks**

The paper addresses the topic of (near) real-time (NRT) data transfer of typical data acquired at a magnetic observatory and discusses challenges and solutions for NRT data transfer from a remote location. The authors discuss the subject applied to the Princess Elisabeth station in Antarctica. In the introduction, the authors present the history, location, and basic operational setup of this research station along with the resulting challenges with respect to operating a magnetic observatory at this location. In the second section of the paper, a definition of NRT data transfer is given, and an overview of traditional data transfer protocols is presented. Subsequently, the authors

discuss possible modern alternative protocols for NRT data transfer. The third section describes the favored MQTT protocol, and the paper concludes with a section on experiences with MQTT and an outlook describing alternative, modern protocols.

NRT data transfer is a recent topic in the operation of magnetic observatories, in particular with respect to space weather and industrial (oil industry) applications. The paper gives an overview of possible solutions, and describes the approach used by the Observatory of Dourbes. In my opinion, this paper will serve as a valuable reference and as a good example how to implement NRT data transfer at a remote observatory. Therefore, I support publication of this article in GI with minor revisions.

Besides the comments of the previous reviewers, in particular the lack of citations, and the lack of detail on the implementation, I have two major suggestions:

First, the language of the paper is often very informal, and sometimes grammatically incorrect. I would strongly recommend to carefully review the grammar and language of the paper. Some exemplary suggestions are given below.

Also, I would suggest to restructure the paper as follows: The first section (Introduction) should not focus on the history and setup of the observatory, but rather introduce the topic as suggested by the title: "real-time data transfer in constrained networks". For example, the definition of near real-time (NRT) (p.2,l.26ff.) should be given in the introduction, and the necessity for NRT at magnetic observatories should be discussed. Also, some examples on previous approaches could be moved to the introduction (p.3, l11-24). Although the application to the observatory should be mentioned in the introduction, I would move the detailed description to section 4, which I would rename "Application to an automated observatory in Antarctica".

**2   Comments on the content of the individual sections**

**2.1   Introduction**

- p.1.l.19: Are six digits the real accuracy for GPS ( 0.11 m)?

- p.1,l.28: Why is a radome used? Any particular reason?

- p.2,l.7: What temperature range are the "extreme temperatures"?

- p.2,l11: ARM processors are primarily used in smartphones, so I guess that's the reason why they are so cheap, and why they are used in Raspberry Pi.

- p.2,l16: Do you use the rugged version? What is its recommended temperature range?

- p.2,l19: Some more details on the self-written software would be appreciated, e.g. are you using GPS time stamps from LEMI/GSM or a separate GPS? What programming language was used? How does the software collaborate with MQTT publisher, e.g. what is the interface to MQTT? Apart from that: If you used the mentioned proprietary file-based windows software of LEMI and GSM, how would you use MQTT in this case –> reading from the file? This might be interesting for Windows users.

**2.2   Towards real time data transfer**

- p.2, l.28: Is this your definition or cited, please specify and cite, if applicable.

- p.3,l.4: Is the delay of 0.3s due to the used datalogging software, or inherent in the LEMI? How did you measure or get this number?

- p.4,l. 16: What does +/- 30 bytes mean here? Does it mean about 30 bytes or XXX bytes +/- 30 bytes?

- p.4 bottom: Shortly describe the exact meaning of "Internet of Things"

**2.3  MQTT explained**

Generally, I would be interested in more details how MQTT can collaborate with existing datalogging software at magnetic observatories, in particular as such software is traditionally based on files. Also, how do you send data to the MQTT publisher in your setting? Such a description may be added to Section 4 as suggested above.

- p.6,l.9-11: I don' understand what "decoupled" means here. You say "a publisher [...] to receive messages." I thought that the subscriber receives messages?

- p.6, l.13: Is there any special meaning of the dashes? E.g., could you subscribe to myhome/floor/room and obtain all sub-messages under this tag?

- p.6,l.16: Are there any security measures, i.e. how is it avoided that everybody on the web can subscribe to your messages?

- p.6, l.17: "are most of the time programs": Except programs, what kind of publishers and subscribers can exist?

- p.6, l.24: How is the description of the data linked to the message? URL? Or is metadata also transferred?

- p.6, l.22: Here, it is not yet clear what the role of the broker is (-> refer to next subsection).

- p.7: Mention which QOS you use and why (can go to section 4).

**2.4 Experiences after one year of data flow**

- p.7, l.23: Setup of MQTT? Please specify what you mean by "setup" here.

- p.7, l.28: Please describe in more detail how MQTT is embedded in your system. For example, you should mention that you use MQTT.js (I assume), and very shortly mention how node.js works. This is related to my general comment in "3) MQTT explained".

- p.7,l.28: You mention that you never close an mqtt connection. However, as I understand it is automatically closed after not receiving messages for the keep alive interval. Please clarify.

- p.8,l.8: You could mention how these 300 ms would compare with fastest feasible rsync / ftp.

- p.8,l.9 After re-establishment of a closed conenction, does MQTT or your program decide when and how data in the memory queue is sent ?

**2.5 Conclusions**

- p.8, l.21f: Does any of these protocols have advantages over MQTT for the described application in Antarctica?

- p.8, l.27: What do you mean by "this" ?

**3 Some exemplary comments how to improve language**

3.1 Abstract and Introduction

- p.1,l.6: Also valid for the remaining paper: Avoid the word "scientists". Better here: In 2013, a project was started by the geophysical data centre at Dourbes to install.....

- p.2,l.9: ... is to [transfer] real-time .... every second [to] Belgium

- p.2,l.12: avoid phrases like "you fight", better: Traditional file transfer protocols (for instance ....) show severe limitations when it comes to real-time data transfer.

- p.2, l.13: Try to remove unnecessary words, like: "on that moment". Another example is p.2,l.28: "and by doing so guaranteeing" can be replaced by "such". Please carefully check for more examples throughout the paper.

- p.2, l.23-26: This sentence is long and a bit confusing.

3.2 Towards real time data transfer

- p.3,l.12: "The one that got our attention is ..." –> "We decided to use...". Try to simplify phrases throughout the paper.

- p.3,l.13: Another example: "...but unlike the latter it is more..." –> "...but is rather..."

- p.3,l.21+22: As we can see –> As shown....

- p.3,l.3: our ARM –> the ARM

- p.3, l.7: "...word will not be possible and to be correct we should always talk..." –> "word is not possible and we should better talk...."

- p.3, l.11: "When we look at the way data transfer is done in the magnetic data collection projects we mostly fall back to file transfer" –> "Usually, magnetic data collection projects use file transfer protocols."

- p.3,l.27+28: "If we need to find....world has to offer": Very informal and a bit confusing. Please rephrase.

- p.4, l.21+22: The phrase: "The days were there were .... are definitely over". may be deleted and "For example," added at beginning of following sentence.

- p.4,l.23+24: "Millions....over the internet" may be removed.

- p.4,l.24: explosion –> increase

- p.5, table: Table number and description are missing.

3.3   MQTT explained

- p.5,l.27: "Message Queue Telemetry is not new. It was designed...." –> "Message Queue Telemetry was designed".

- p.6, l.2: "In 2011,....seconds.  What makes....us that in 2013 IBM...."  may be replaced by "In 2013, IBM...."

- p.6,l.28: Please rephrase "Also here....available"

- p.7,l.6: Usage of "Ok" is very informal. Better: "is acceptable"

---

## Author Comment (AC3) · 28 Apr 2017

First of all thanks for this detailed review with lots of interesting remarks. It took me a couple of days to update the paper and include most of the remarks made in this review. Special attention was given to the language used and the restructuring of the document. It's available as attachment. I only include here some comments on things I didn't include in the new paper and the reasons why. a) Some more details on the self-written software would be appreciated,how MQTT is embedded in your system. For example, you should mention that you use MQTT.js (I assume), and very shortly mention how node.js works. MQTT is just a specification that describes a protocol, to use it you need to include a library in the programming language of your choice. This library

will always have basic mqtt methods: connect, deconnect, subscribe,unsubscribe, publish. On the site of mqtt https://github.com/mqtt/mqtt.github.io/wiki/libraries you find over 60 libraries for different programming languages. I use currently three programming languages C#( windows autodif program ) javascript ( because of nodejs ), and python all with different libraries to publish/subscribe to mqtt topics, but with similar approaches to use MQTT. The choice of using nodejs has nothing to do with the use of MQTT. After 15 years of software engineering experience on building websites with most of the time java solutions ( still one of my favourite programming languages), nodejs was the first time that I saw a lightweight asynchronuous event-driven approach to web development which scaled and preformed well. Node.js changed the way web sites are build today compared to the more classical approaches (Java EE,ASP.NET, Ruby ON rails) It was also perfect to install on beaglebones and build a web interface. The ideas of node.js are currently integrated in python 3.5 with asyncio libraries and webframeworks as Sanic or on the database site asyncpg. Although these are very interesting topics to discuss on, I think they are out of scope for this paper. b) If you used the mentioned proprietary file-based windows software of LEMI and GSM, how would you use MQTT in this case –> reading from the file ? This might be interesting for Windows users . Reading the file and using MQTT to send the last added second/minute would not add any value compared to the classical ways of transferring data ( Rsync,FTP etc) but can be done. Software changes are needed to embed MQTT and send at the same moment as the writing to the file.All software I written works on windows as well.

Please also note the supplement to this comment:
http://www.geosci-instrum-method-data-syst-discuss.net/gi-2017-17/gi-2017-17-AC3-supplement.pdf

---

## Referee Comment (RC4) · S. Flower (Referee) · 5 May 2017

General Comments

This is a useful paper for anyone evaluating this type of communication problem (reliable data delivery from a remote location over an unreliable network) and is not limited to the Geomagnetism community. It summarises well the decisions that the authors made and gives clear details on how to implement a similar solution.

Specific Comments

The 56kb/s that you have available for your WAN link is a large bandwidth for geomagnetic data. It equates to about 5000 bytes per second of uncompressed data (when

using 8 bit data with 2 framing bits and 1 parity bit). You are transmitting 16 bytes / second from the vector magnetometer and 8 bytes from your scalar magnetometer, so you have plenty of bandwidth overhead (suing less than 10% of available bandwidth).

Another option that you don't list is to write your own TCP or UDP protocol. This is not complex with modern languages access to the Sockets library (for example). Did you consider this alternative?

Allowing for loss of network connection of up to 4 hours - is this enough? I see you recovered a longer period than this using ftp, so you have some sort of backup file store on the data logger that you can acces remotely? When we have lost network connections, the time taken to recover can be in days or even weeks. With the low cost of memory, would it be posible to introduce a much larger memory buffer into the MQTT transfer system? If this large buffer were ever to be needed, does MQTT allow for prioritisation of current data over older data (some seismc data loggers protocols allow for this)?

Technical Corrections

Some line by line typographical corrections / suggestions.

Page 1: Line 8: "values *up* to" should read "values down to" Line 9: "lemi-25" should read "lemi-25 vector magnetometer" Line 10: "gems proton magnetometer" should read "GEM Systems scalar proton magnetometer" Line 11: "DI-flux" should read "DI-fluxgate magnetometer" Line 12: "real-time" should read "real-time capability" Line 14: What is UGCS? Line 16 "it and arrives at" should read "and it arrives" Line 25 "profit of" should read "profit from" Line 27: "meters of" should read "meters from" Line 28: by "pillars" do you mean the pillars used for geomagnetic measurement?

Page 2: Line 2: Did you have a power budget? Line 6: "chose for the" should read "chose for this the" Line 8: "GEMS" should read "GEM Systems" Line 15: Is it well documented? This would be neccessary to make adaptations. Line 20: What has the

software been re-written from? Was there a Windows version of the software that had to be re-written?

Page 3: Line 8: "data capturing on the serial port" should read "data logger's serial port" Line 12: "this one" should read "this" Line 14: "differences" should read "differences between two files" Line 26: "where" should read "were": "on" should read "to"

Page 4: Line 2 "earth" should read "Earth's": "a shared knowledge" might be better as "shared techniques" Line 4: I think Antelope software is created by Boulder Real Time Technologies and sold through Kinemetrics. USGS are only one user of this software. If you are mentioning SeisComp3, you should credit GFZ where the software originates. Line 19: I've used Earthworm and found the support to be excellent. The commercial software providers, Instrumental Software Technologies Inc, even did some work for us without asking for payment. Line 24: "scientist" should read "scientists" Line 25: "profit of" should read "profit from" Line 26: "on remote" should read "at remote"

Page 5: Line 9: Isn't MQTT and ISO standard (ISO/IEC 20922:2016)? Line 11: "While taken the definition from there respectively website" doesn't make sense. Do you mean "Here are the definitions of each of these protocols from the relevant websites". Can you add the website links to your list of references? Line 21: "my" should read "our"

Page 6: Line 3: "from several seconds to hundreds of seconds" - this means the real-time performance got worse! Line 14: "on regular intervals a temperature" should read "at regular intervals a thermometer" Line 24: "This topic is also linked with a description of the message 25 structure posted on it" - what does this mean? Does MQTT include metadata that describes the topic? Line 27: "before the" should read "before they" Line 29: "these server can" should read "of the server to"

Page 7: Line 7: "what the possibility of losing messages during transfer is" should read "the possibility of losing messages during transfer" Line 9: "send" should read "sent": "It will be doing so never introduce duplicates" - doesn't make sense? Line 11: "send" should read "sent" Line 14: "send" should read "sent"

[Figure]

Page 8: Line 28: "realisation in" should read "realisation of"

---

## Author Comment (AC4) · 10 May 2017

Thanks for all comments,

General comments on 56 kbits/sec : You are probably right on the fact that I still have a lot of extra bandwidth with my first message structures being 8 bytes and 16 bytes , but in first implementation I pushed it to the limit so I had these small messages. In current versions I have already bigger messages but still small where I pass also temperature info ( 24 bytes vector and 12 bytes for scalar) Althouhg it still is small extra bandwidth can come in handy when we get bursts of messages whenever connections got stale or lost. General comments on considering to write my own protocol ? I never considered to write it myself, because when I would do it, it would never be something

like a message oriented middleware (MOM) with a broker that comes with complete the decoupling ofpublisher/subscriber but rather be a request/reply protocol. Also writing the code would mean for me writing it again in for the moment three languages (C#,javascript,python) and maintaining/testing it in these three languages ( and maybe more ). The code of mqtt publisher/subscriber is simple and easy to integrate in most popular languages. The complexity lies in the broker which is done for you (it comes with a configuration and management effort). the liberty of adding different subscribers to the same topics doing different obs (one for database insertion, one for alerts on different patterns: gps failures, temperature etc) opens up different possibilities without messing up code.The instruments now default always log to file, by enabling mqtt they can send data to topics without change default behaviours.

Allowing loss of network connection for 4 hours is this enough ? As I stated already I indeed log data locally to files , this is a must because in the unmanned observatory it can be that connection of satellite is lost and only can be switched back on in the next summer. When I began to use the library mqtt.js in 2014 this library was doing the retransmit of messages itself even when connection was lost ( not in mqtt standard ), however doing this it stored the messages in memory while testing two days of connection failure it got an out of memory. I adapted this code by rewriting handlers so I can now limit the messages that it will store ( and is configurable in a file for each topic). I looked at it today and see that the config is currently 1 hour for second data and 4 hours for minute data. In the current version of this api they make a storage available and you can use levelDB ( key value nosql database) to store the not sended messages which now gives the opportunity to go way beyond several days (also not standardized and not necesssary in api). But then again after some reasoning of what exactly should be done to use mqtt probably one day is the maximum ( the real-time aspect is already lost we only use it to deliver every message, for me one hour was sufficient ) . To recover more then this I fallback to ftp approaches because the data is available at the antarctica site. There is another aspect that also becomes important on this level and it is the scalability of your broker. Having a burst of one week second

data needs to scale well and messsages need to be consumed quickly. Each topic can be limited to a maximum of queued messages,etc .This all depends on the choice of broker. The broker I use is not the best on scalability but then again I didn't tested it for burst more then four hours. Does MQTT allow for prioritisation ? No, MQTT doesn't have prioritisation in his standard , it is available in AMQP (which also has message routing and etc) but this protocol is not lightweight. You could e.g. use another topic for messages that were not sended during a long connection loss while not influencing the real-time topics.

Technical Corrections Most of the technical corrections are included in the new version. Some extra notes : Line 15: Is it well documented ? All technical shemes can be found on https://github.com/beagleboard/BeagleBone-Black and http://www.elinux.org/Beagleboard:BeagleBoneBlack#Hardware_Files Line 20: What has the software been re-written from? Was there a Windows version of the software that had to be re-written? the rewritten is true for the lemi-025 I rewrote everything that was possible in the windows by studying the user interface and remake it into a web interface (possibility to see current measurements, update dacs, stop/start measurements, change parameters ,etc) Isn't MQTT and ISO standard (ISO/IEC 20922:2016)? yes it is, I didn't know it and added it in the paper. Line 24:What does this mean ? Does MQTT include metadata that describes the topic ? I rewrote this sectionbecause it was not clear. MQTT doesn't include metadata it is up to the analyst/developer to document message structures ( this can be binary formats ( what I use ) or text ( JSON if we want but messages will be bigger in this case )) there is no WSDL like in soap webservices. Best regards, Stephan